# Development of Master Chef: A Curriculum to Promote Nutrition and Mindful Eating among College Students

**DOI:** 10.3390/ijerph21040487

**Published:** 2024-04-16

**Authors:** Kayla Parsons, Kelley Strout, Caitlyn Winn, Mona Therrien-Genest, Kate Yerxa, Jade McNamara

**Affiliations:** 1School of Food and Agriculture, University of Maine, Orono, ME 04469, USA; kayla.l.parsons@maine.edu (K.P.); caitlyn.winn@maine.edu (C.W.); mona.therrien@maine.edu (M.T.-G.); kate.yerxa@maine.edu (K.Y.); 2School of Nursing, University of Maine, Orono, ME 04469, USA; kelley.strout@maine.edu

**Keywords:** mindful eating, curriculum development, college students, diet quality

## Abstract

Research suggests that success in improving undergraduates’ diet quality can benefit from a multifaceted approach, incorporating nutrition education, mindful eating, and culinary skill-building. The current study aimed (1) to review the development of Master Chef, a mindful eating curriculum, and (2) assess its feasibility through an online expert review. Expert reviewers were recruited through an online mindful eating course. Survey questions included both Likert-style and open-ended questions. Quantitative survey data were analyzed using descriptive statistics. Two independent researchers coded qualitative data, which then underwent inductive thematic analysis. Reviewers (N = 7) were experts in the fields of nutrition, psychology, and mindful eating. Master Chef’s overall feasibility was rated highly. The overall curriculum was perceived positively. However, it was recommended that the program include more mindfulness. Master Chef was identified as a feasible program for improving the health behaviors of college students. Pilot dissemination and analysis will be necessary to assess the program’s effectiveness in supporting disease prevention among undergraduates.

## 1. Introduction

Young adults experience a transitional period when entering college, which influences their health behaviors. This transitional time in their lives is often impacted by personal, social, and environmental pressures [1]. In addition, college students have notably poor diet quality as a result of inadequate consumption of whole foods and excessive intake of saturated fat, sodium, and calories [2]. Personal knowledge and skills, social influence, the college food environment, access to healthful foods, and perceived stress influence these dietary choices [3]. Poor dietary choices and lifestyle behaviors during college often continue post-graduation, further increasing the risk for diet-related chronic diseases, such as cardiovascular disease and diabetes mellitus [4]. To mitigate the development of chronic disease, implementation of college-based interventions has aimed to improve diet quality through culinary skill-building and nutrition education.

Szczepanski et al. developed Culinary Boot Camp, a college-based nutrition education program that aimed to improve diet quality and nutrition knowledge [5]. The four-week curriculum provided upperclassmen with a 30 min nutrition lesson, an hour-long cooking experience, and a 20 min group mealtime. After completing the curriculum, students had significantly improved in their cooking skills, cooking attitudes, and healthy eating self-efficacy. In addition, students had significantly improved their dietary intake of vitamin C, magnesium, potassium, and fiber [5]. McMullen et al. identified similar findings after implementation of College CHEF [6]. The curriculum lessons were offered weekly for four weeks, with each two-hour lesson addressing specific culinary techniques while emphasizing the use of fruits, vegetables, and seasonings when cooking. Post-intervention, participants had significantly increased levels of confidence in using fruits, vegetables, and seasonings when cooking as compared to controls; however, no assessment of diet quality was provided. Clifford et al. conducted a randomized control trial assessing the impact and feasibility of a recorded cooking show, Good Grubbin’, on college students’ health behaviors [7]. This short intervention consisted of four 15 min videos, which resulted in significant improvements in knowledge surrounding fruit and vegetable guidelines within the intervention group [6]. These curricula provide evidence for incorporating culinary aspects into health promotion interventions designated for college students but lack the ability to address additional influencers of dietary choices, such as poor mental health, in this population [8,9].

Researchers have revealed that poor mental health negatively influences dietary choices and one’s relationship with food. College students are at a heightened risk of disordered eating, such as binge eating and dieting, which can be partially attributed to the lack of coping strategies despite high levels of stress [10]. Eating behaviors, such as periodic excess consumption of calories and weight cycling, exacerbate the risk of cardiometabolic disease [11]. Research has suggested the potential role of mindfulness, the non-judgmental awareness of moment-by-moment experiences, in cultivating stress management [12]. Mindful eating, the act of noting both internal and external cues during eating, has the propensity to act as a bridge between positive stress-coping behaviors and healthy relationships with food [13]. This may be partly due to the construct’s emphasis on self-compassion [14]. Practicing mindful eating allows for intention in dietary choices while navigating external influences on the diet, including appearance or aesthetic-driven influences [13]. This subsequently promotes body appreciation, the ability to accept and hold favorable attitudes towards one’s own body, while rejecting societal beauty norms [13]. In a similar fashion to practicing mindful eating, higher levels of body appreciation have been associated with reduced eating psychopathology [15].

There have been few college-based attempts at promoting nutrition and stress management using mindful eating, with researchers, instead, targeting populations with obesity [16]. Mindful Eating and Living, or MEAL, was an intervention that offered two-hour weekly sessions containing meditation, group eating sessions, and basic nutrition information, which were offered to women with a BMI of 30 kg/m^2^ or more. Mindfulness, eating behaviors, and inflammatory markers were measured at baseline, after intervention completion, and during a three-month follow-up [16]. All participants had lost weight at the follow-up compared to baseline. Frequency of binge eating, levels of c-reactive protein, and mindfulness significantly improved immediately after the intervention and at the three-month follow-up [16]. This research supports the blending of mindful eating and health promotion. In addition, short mindful eating interventions have been found to be effective in promoting eating behaviors [17]. Allirot et al. compared study participants’ food choices after they viewed a seven-minute mindfulness video or an unrelated video of the same length [17]. After finishing the video, all participants were provided with finger foods of varying nutrient and caloric density. Interestingly, those in the experimental mindfulness group ate significantly less of the higher-caloric items and less dietary fat than the control group, although the number of finger foods consumed did not significantly differ [17]. This finding suggests that even brief mindfulness education can have a positive short-term impact on participants’ food consumption [7]. Knol et al. had similar findings after the implementation of a mindful eating culinary program for undergraduate, graduate, and medical students [18]. Mindful eating ability significantly improved at follow-up as compared to controls [18]. Several subdomains of mindful eating, including disinhibition, eating beyond satiety, and eating with awareness, were also improved at follow-up when compared to the control group. This research suggests that undergraduates can benefit from a multidimensional approach, incorporating mindful eating, culinary skill-building, and nutrition literacy [19]. More research is needed in developing a curriculum that builds on previous efforts. Additionally, programs require formative analysis that ensures program viability and validity [19].

Based on previous research on the relationship between mindfulness and eating behavior change, Master Chef was developed. This curriculum incorporates experiential learning centered on culinary experiences, nutrition education, and body appreciation through mindful eating among college students. The current study aimed to review the development of Master Chef, in addition to assessing its feasibility through expert review.

## 2. Materials and Methods

### 2.1. Study Design

The Master Chef curriculum was developed to address college students’ nutritional needs and perceived barriers to healthful eating while integrating mindful eating practices. Specific issues cited by college students in the previous literature included poor fruit and vegetable intake, time constraints, lack of access to healthful food, and poor mental health [3,20,21]. The curriculum was reviewed by experts from nutrition, psychology, and mindful eating and then refined. Reviewers provided qualitative and quantitative feedback on the curriculum’s educational content, lesson objectives, and perceived feasibility. This research was approved by the University of Maine’s institutional review board, and the expert review was conducted in the spring of 2023. This study was conducted in accordance with the Declaration of Helsinki, and all subjects provided informed consent for inclusion before participation in this study.

### 2.2. Curriculum Development and Theoretical Support

The curriculum was developed through the lens of the social cognitive theory to promote evidence-based behavior change. Each of the four lessons adhered to the construct of reciprocal determinism, emphasizing the interaction between environmental factors, cognitive factors, and intrapersonal behaviors [1]. Participants are provided with health-promoting education, followed by discussions surrounding the application of knowledge in a college environment. Participants can then customize recipes, addressing key intrapersonal factors, such as taste and cooking self-efficacy, in an experiential learning environment. Social cognitive theory has previously been used successfully to improve health behaviors through multiple determinants, including knowledge, perceived self-efficacy, outcome expectations, goal-setting, and perceived facilitators within their environment [1]. The curriculum emphasizes the role of diet quality in preventing chronic disease and mindful eating in navigating food choices. Healthy eating self-efficacy is promoted through the experiential learning process. All recipes chosen for the curriculum are low-cost, with ingredients that are easily accessible; furthermore, recipes are designed to be prepared with limited cooking appliances and utensils, taking into consideration students’ environments. Students are then encouraged to complete specific, measurable, attainable, relevant, and time-bound (SMART) goals through behavior-based homework assignments directly related to coinciding lesson topics.

Each lesson is approximately 1.5 h long. Each lesson includes nutrition education, mindful eating, and body acceptance education, followed by a culinary experience in which students create a nutrient-dense recipe. At the beginning of the program, a cookbook developed by a registered dietitian nutritionist, containing accessible, ‘dorm-friendly’ recipes is provided. During the lessons, groups of approximately 4–5 participants are supplied with the necessary ingredients to create one recipe per lesson. To encourage creativity and cooking self-efficacy, students are provided with additional ingredients to customize their recipes. After completion, participants share their meals and practice mindful eating in a group setting. This is followed by a discussion surrounding the mindful eating experience, how to apply key lesson topics in their life, and future application post-graduation. Participants also receive take-home food bags with the necessary ingredients to create the lesson’s recipe in their dorms. See Table 1 for further detail.

### 2.3. Expert Review Recruitment

We employed purposeful sampling by recruiting potential reviewers enrolled in an online mindful eating certificate program for health professionals, separate from the current study. We gained approval from the course administrators to post a survey link for the expert review on the course’s discussion board. A total of 110 individuals were enrolled in the mindful eating program. These individuals had equal and unlimited access to the course’s discussion board. It should be noted that individuals enrolled in the online mindful eating course were not required to participate in the current study as a part of the course. The goal was to recruit 20 reviewers, based on previous research recommendations [22]. Expert reviewers received a USD 75 honorarium after completing an online survey.

### 2.4. Expert Review Survey

The expert review survey contained access to the consent form, the entire Master Chef curriculum, and corresponding supplemental material. Reviewers were instructed to read the entire curriculum and supplemental material before answering questions corresponding to each respective lesson. Respondents were also asked to provide feedback on the overall concept of the curriculum. Survey questions were both open-ended and Likert-style, including reviewers’ ranking of course acceptability on a scale of 0–10, with higher scores representing increased acceptability. Participants were included in the analysis if they had consented to participate, and they used the expert review survey link to provide feedback on all sections. Refer to Table 2 for further details regarding the survey questions. Participants had the opportunity to leave the survey webpage and come back as convenient for them.

### 2.5. Analysis

Survey questions were analyzed using descriptive statistics and/or thematic analysis, based on question type. Quantitative data were analyzed in Microsoft^®^ Excel^®^ for Mac, Version 16.81 (Microsoft^®^ Excel^®^ for Mac, Released 2024., Version 16.81, Albuquerque, NM, USA, © 2024 Microsoft) using central tendency methods to assess the mean and median for continuous data, such as years of working in field. Frequency distributions were used for categorical data, such as highest level of education and area of work. This analysis was completed using the International Business Machines (IBM) Statistical Package for Social Sciences (SPSS) Statistics for Windows Version 28.0 (IBM Corp., Released 2021, SPSS Statistics for Windows, Version 27.0. Armonk, NY, USA: IBM Corp). Open-ended questions were assessed by two independent researchers, using inductive thematic analysis [3]. Using this method, we read the survey results repeatedly to become familiar with the qualitative data. Based on the survey results, we developed initial codes representative of key features throughout all responses. Based on coding, we independently developed broader, latent themes that encompassed corresponding codes [23]. Themes were data-driven, rather than using a pre-existing coding frame [23]. The individual members of our research team then collaborated and revised the themes, producing the final results [23]. Please refer to Figure 1 for an overview of the methods.

## 3. Results

### 3.1. Expert Review Sample

A total of 110 experts were enrolled in the mindful eating certificate course and had access to the online discussion board, which contained a recruitment link, consent form, and corresponding expert review survey. Out of the 110 who had access to the discussion board, 16 individuals clicked on the link to enter the online survey. A total of seven expert reviewers had complete datasets, which were used for analysis. On average, it took the reviewers 11.9 ± 25.3 h to complete the review process. The expert reviewers (n = 7) were primarily registered dietitians (n = 4), followed by a professor (n = 1), psychotherapist (n = 1), and nutritionist (n = 1). Most reviewers had a bachelor’s degree (n = 3, 42.9%) or a master’s degree (n = 3, 42.9%), and one participant had a doctoral/terminal degree. The average years spent in their respective fields were 21.0 ± 7.5 years (ranging 6–30 years).

### 3.2. Expert Reviewers’ Perceived Program Benefits and Feasibility

Reviewers stated that the program would improve health behaviors, including diet quality (M = 8.5 ± 1.6), cooking skills (M = 8.7 ± 1.5), cooking self-efficacy (M = 9.0 ± 0.6), and body appreciation (M = 7.7 ± 3.0). The overall feasibility of the curriculum was also ranked highly (M = 8.5 ± 1.6).

### 3.3. Expert Reviewers’ Perception of Master Chef Lessons and Overall Program

Lesson One: Mindful Eating and Nourish Bowls. In this lesson, participants are introduced to the concept of mindfulness and mindful eating and are provided with nutrition education describing strategies to incorporate fruits and vegetables. Two themes were identified in this lesson: (1) modifying the mindful eating activity, and (2) attention to detail. Reviewers suggested modifying the mindful eating activity by incorporating the practice of mindfully eating a raisin. For context, this popular practice emphasizes experiencing all senses while eating a raisin and assessing how your body reacts to the experience. Other suggestions included being less structured regarding mindful eating, such as not quantifying the number of chews per bite. Reviewers described the lesson as detailed, with one reviewer stating, “I found this to be very clear, detailed, and was not confused at all. All references and extras were in the appendix section. Colorful recipes and infographics were extremely helpful”.

Lesson Two: Non-Judgement and Redefined Ramen. In this lesson, participants are asked to identify potential unmet needs based on thoughts that occurred while practicing mindful eating, followed by a body appreciation thought exercise; in addition, participants are taught about the health benefits of incorporating plant-based proteins. Two themes were identified within lesson two. The first theme is the creative application of mindfulness, with one reviewer stating, “I like the addition of incorporating the different facets of mindful eating in an experiential fashion”. The second commonly discussed theme of this lesson is attention to detail.

Lesson Three: Food Choices and Overnight Oats. During lesson three, participants are encouraged to assess influencers of their dietary choices (in regard to potential environmental and personal influences). Participants then engage in a brief discussion on social determinants of health, and approachable solutions to barriers of eating healthfully. Macronutrients and micronutrients are also described during this lesson, emphasizing making dietary choices to meet the body’s needs best. Themes identified from the expert review include (1) the lesson being detailed, (2) it being informative, and (3) suggested modifications to the mindful eating activity. Reviewers enjoyed the education surrounding nutrients and praised how the lesson was organized, with one reviewer stating, “As are previous sections, this is detailed and compassionate. I enjoy the group interaction”.

Lesson Four: Food Origins and Meal Prepping. Participants are provided with a locally grown food (blueberries) and a gummy worm during this lesson. Participants are introduced to food systems through a discussion of contrasting food origins, emphasizing choosing whole foods. Participants also reflect on college-specific barriers that influence food intake, to help improve diet quality and encourage meal prepping. Reviewers again described this lesson as detailed but provided several suggestions for modification of the lesson plan. One reviewer recommended incorporating more of a discussion on planetary health in the lesson. Another reviewer stated that more care should be used when discussing dietary choices, stating, “There is a fine line when discussing food choices and mindfulness”.

Overall Opinion of the Curriculum. Most reviewers positively perceived the overall curriculum. All the reviewers highlighted the curriculum as being organized and detailed. There were frequent comments regarding the need to convey the role of mindfulness and meditation to participants throughout the curriculum. One comment exemplifying this theme was, “Missing from the curriculum is any mention of the value of (some would say necessity) of having mindfulness practice in addition to eating mindfully. Most people will be able to practice mindful eating better/easier if they also have a meditation practice…even if only for 10 min”. Refer to Table 3 for a summary of themes for the overall curriculum with supporting quotes.

## 4. Discussion

### 4.1. Discussion Overview

The current study aimed to review the development of a mindful eating curriculum, Master Chef, which leads experiential learning through culinary experiences, nutrition education, and body appreciation, all centered on mindful eating. The curriculum was developed using social cognitive theory to address barriers to healthy eating among college students [1]. The curriculum underwent formative evaluation by way of an expert review and was positively perceived by reviewers, with moderate modifications, including incorporating mindfulness practices. It should be noted that formative evaluation of nutrition-based curricula through an expert review is rare in the literature. The current findings add to the few previously published, illustrating the benefits of utilizing a panel of expert reviewers for curriculum refinement, leading to improved content validity and feasibility.

This method of formative assessment has primarily been utilized within nutrition programming developed for adolescents; however, the current study is unique in using this technique for assessing feasibility of college-based programming [24,25,26]. Despite population differences, both Master Chef and previous curricula targeting multiple health behaviors have been viewed as feasible by academics and health experts [24,25,26]. Project Stride, an enrichment program delivering nutrition education to traditionally marginalized youth, through the lens of science, mathematics, and art, was reviewed by community partners and education specialists, including community nutrition educators and elementary school teachers. Similarly to Master Chef, the reviewers felt confident that it was a program they could teach to the targeted audience [24]. Likewise, one Master Chef expert reviewer stated, “Absolutely wonderful curriculum! I plan to use components in my own mindfulness classes”. Additionally, Teens CAN: Comprehensive Food Literacy in Agriculture and Nutrition addresses similar lesson topics to Master Chef and was found to be feasible by Cooperative Extension experts [25]. Regarding content validity, expert reviewers of Master Chef agreed that the program has the potential to improve diet quality, cooking skills, and self-efficacy, as well as body appreciation. Likewise, expert reviewers of the nutrition education and food safety program Eat Right: Future Bright acknowledged that objectives were being met within the lessons, leading to the promotion of nutrition, physical health, and food safety [26].

Emerging themes from the Master Chef expert review included creative application of lessons, minor modifications for activities, and a noted attention to detail. Similar themes have been identified in previous curricula as a result of formative assessment [24,25,26,27]. Master Chef expert reviewers viewed the experiential learning as engaging and a contributing factor to promoting social health, coinciding with previous research implementing this learning strategy [24,25,26,27]. These findings are similar to those for the In the Defense of Food curriculum, in that expert reviewers agreed that the incorporation of multiple learning strategies, including hands-on or kinesthetic methods, can promote independent thinking and learner accessibility [27]. Coinciding with previous curricula, Master Chef expert reviewers discussed the need for modifications of course activities to improve feasibility among students. [24,25,26,27,28]. In Project Stride, for example, reviewers underscored the necessity of modifying nutrition lessons to be culturally informed, time-sensitive, and age-appropriate [24]. Additionally, Master Chef and other curricula expert reviewers have consistently identified organization as an essential component in supporting program implementation and replicability [28]. The need for supplemental material, a lack of lesson scripts, and poor time management have been previously cited as major areas of improvement in program refinement [24,28]. However, expert reviewers described Master Chef as being organized and helpful to future educators due to its curriculum presentation, illustrations, and the provision of in-depth supplemental material. Reviewers described the Eat Right, Future Bright curriculum in a similar fashion [26].

Despite previous efforts being found effective within adolescent programming, there have been limited attempts at assessing college-aimed programming through the use of expert review. Undergraduates uniquely face alarming levels of perceived stress and poor diet quality, requiring theory-based programming that promotes self-efficacy to overcome environmental barriers [1,2]. Master Chef effectively addresses this, as determined by experts, through providing relevant tools for students (nutrition education, culinary skills, and mindfulness) and being detail-oriented in nature.

### 4.2. Curriculum Refinement and Future Implementation

Expert reviewers had several suggestions regarding the mindful eating practice activities and mindfulness content. The practice of mindfulness has a foundational role in mindful eating and influences operant conditioning by regulating reward-based learning systems [29]. When mindfulness practices are incorporated into the act of eating, individuals experience decreased expected award and pleasure from food while also being able to identify influences of dietary behaviors with a non-judgmental approach [29]. Due to the complex nature of the college environment, an interdisciplinary approach, led by experts in their respective fields, is required to meet students’ needs. Master Chef will be implemented as part of WellNurse: a Holistic Multidimensional Intervention [30]. WellNurse aims to address systematic burnout and increase resilience among baccalaureate nursing students. This intervention is interdisciplinary, with experts in their respective fields leading initiatives in mindfulness, such as Mindfulness-Based Stress Reduction, mindful physical activity, mindful eating, and nutrition education, alongside a system-wide promotion of a culture that exhibits resilience and community [30]. The curriculum will be delivered during a one-week research learning experience prior to the beginning of the fall semester.

### 4.3. Limitations

Several limitations arose during the assessment of the curriculum. There was a failure to reach the goal recruitment rate of 20; however, based on the reviews, we surmise that saturation was met through thematic analysis [31,32]. Furthermore, we conducted purposeful sampling through recruitment of experts in the field of mindful eating, in addition to other relevant fields (i.e., psychology, nutrition). This provided a rare in-depth context regarding the feasibility of the program, despite low recruitment [31]. Of note, similar program assessments, as conducted with Project Stride, Eat Right, Future Bright, and Eating Smart: Being Active, have had comparable sample sizes, ranging from five to nine participants when using expert review [24,26,28]. These findings infer that an increased sample size would not add additional insight for program refinement due to meeting saturation with the current sample. However, the quantified program feasibility and the perception of the program as improving health behaviors may have been influenced by the limited number of expert reviewers, with more reviewers potentially leading to greater variance within the dataset.

It should be noted that 16 individuals attempted to complete the expert review, but several failed to produce complete datasets. This suggests that individuals enrolled in the mindful eating course, with most working in healthcare or academia, have limited time to conduct an expert review. The average time to complete the review was extensive and, therefore, may limit the feasibility of future curricula studies following similar methods of a rigorous review process. Moreover, the respondents were unequally distributed among professions, with the majority being registered dietitian nutritionists. This may have influenced the themes, including perceived feasibility, which may have appeared differently if the reviewers had previous experiences with providing nutrition education. The Master Chef curriculum needs to undergo validation testing to assess the efficacy of using this curriculum to change mindful eating behavior. This will allow us to evaluate improvements in body appreciation, self-efficacy in healthy eating, nutrition literacy, and total diet quality in undergraduate students after participating in Master Chef.

## 5. Conclusions

Master Chef is a short, theory-based mindful eating culinary intervention designed for college students, which integrates multiple influencers of diet quality within a population at risk of facing adverse health outcomes. Included in the curriculum are the promotion of culinary skill self-efficacy, nutrition literacy, body appreciation, and mindful eating while also addressing potential limiting factors of the college environment itself. Expert reviewers evaluated the curriculum as feasible with the potential to greatly improve body appreciation, cooking self-efficacy, cooking skills, and overall diet quality. Pilot dissemination and a corresponding analysis are necessary to grasp the overall effectiveness of the program in supporting disease prevention among undergraduates.

## Figures and Tables

**Figure 1 ijerph-21-00487-f001:**
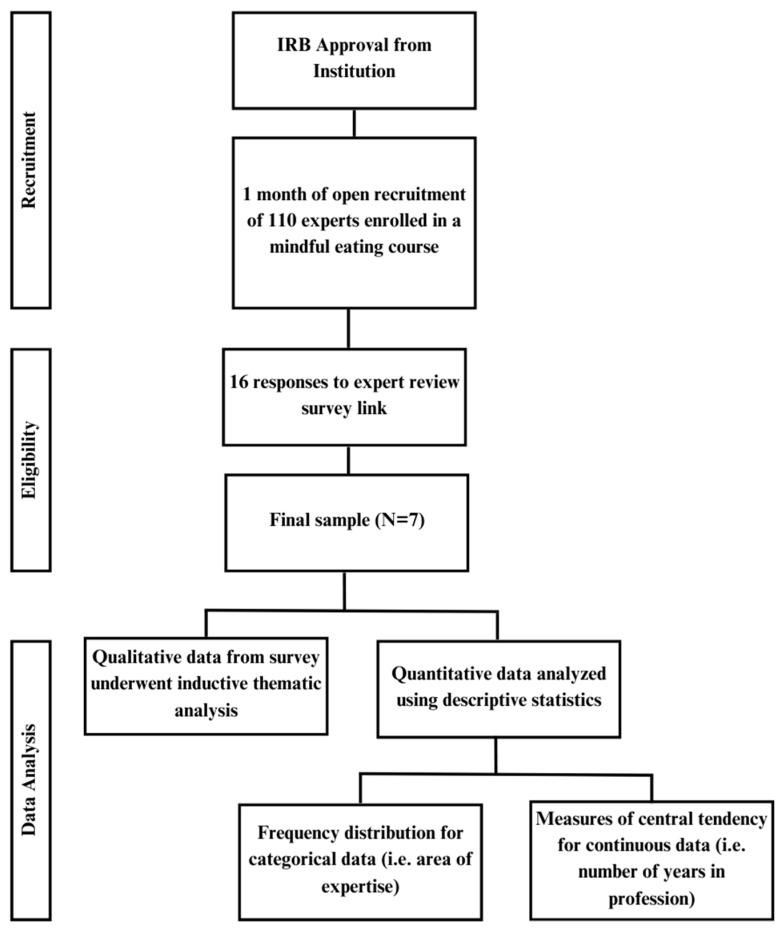
Overview of methods for Master Chef expert review.

**Table 1 ijerph-21-00487-t001:** Master Chef lesson objectives and content overview.

Lesson Name	Lesson Objectives	Lesson Content
Mindful Eating and Nourish Bowls	Demonstrate the ability to practice basic mindful eating skills as measured by completion of the mindfulness scoreboard by the end of the lesson. Successfully create a recipe that incorporates fruits and/or vegetables by the end of the lesson. Describe at least one strategy on how to incorporate eating more fruits and vegetables into their diet during the discussion at the end of the lesson.	Adding fruits and vegetables into a busy schedule 4 Principles of Mindful Eating
Non-Judgment and Redefined Ramen	Demonstrate the ability to practice basic mindful eating skills as measured by completion of the mindfulness scoreboard by the end of the lesson. Identify one strategy to promote self-compassion by the end of the lesson, as measured by post-class discussion. Successfully create a plant-based recipe at the end of the lesson. Describe how to incorporate one plant-based food into their diet by the end of the lesson, as measured through the post-class discussion.	Plant-based proteins Identifying thoughts while mindfully eating ‘Freedom from Blame’ exercise
Food Choices and Overnight Oats	Demonstrate the ability to practice basic mindful eating skills as measured by completion of the mindfulness scoreboard by the end of the lesson. Identify influences of their dietary choices and develop plans to promote healthful eating in response to barriers by the end of the lesson, as measured by the in-class activity. Successfully follow a meal-prep-friendly recipe by the end of the lesson.	Introduction to macronutrients and micronutrients Eating for physical and mental needs Barriers to mindful eating in college Social determinants of health
Food Origins and Meal Prepping	Demonstrate the ability to practice basic mindful eating skills as measured by completion of the mindfulness scoreboard by the end of the lesson. Identify benefits of meal prepping and effective ways to meal prep, as measured by the discussion. Successfully follow a meal-prep-friendly recipe by the end of the lesson. Discuss where ingredients from their favorite dishes come from by the end of the lesson, as measured by the discussion.	Introduction to food systems Choosing whole foods over ultra-processed foods Benefits of meal prepping and food safety

**Table 2 ijerph-21-00487-t002:** Expert review online survey questions.

Topic	Questions
Sociodemographic	Which category or categories best describes your area of work? What is your highest level of education? Please list any additional credentials below. For how many years have you been working in your field?
Introduction	Were there any areas that were confusing when reading through the introduction of the program? Please explain. What additional information should be included in the introduction?
Individual Lessons ^a^ (1–4)	Are the learning objectives for the lesson clear and adequately addressed throughout the lesson? Are the lesson content and culinary application appropriate and do they occur in sequential order? Please provide feedback on the lesson: likes, dislikes, areas of confusion, etc.
Overall Curriculum	After reading the curriculum, how confident do you feel that you could *teach* the program, on a scale of 0–10? What was difficult to understand about the curriculum, overall? What was easy to understand about the curriculum overall? Does the overview of the curriculum explain the scope and materials needed to run Master Chef? Please explain. Are the lesson content and culinary application appropriate and do they occur in sequential order?
Feasibility and Behavior Change ^b^	What is the feasibility of running the curriculum on your college campus? Would this curriculum improve the diet quality of college students? Would this curriculum improve the cooking skills of college students? Would this curriculum improve the cooking self-efficacy of college students? Would this curriculum improve body appreciation among college students? Please provide any additional feedback or comments.

^a^: Questions were repeated in separate blocks for assessment of each corresponding lesson. ^b^: Preceding each question was “On a scale of 0–10…”.

**Table 3 ijerph-21-00487-t003:** Themes and supporting quotes from expert review (N = 7) thematic analysis.

Theme	Supportive Quotes
Positively Perceived	“This was fabulous! I want to do it! It really was exciting to review it and appreciate the opportunity to give feedback. I truly have no negative feedback or suggestions.” “I found this to be very clear, detailed, and was not confused at all. All references and extras were in the appendix section. Colorful recipes and infographics were extremely helpful.”
Need to Incorporate More Mindfulness	“Missing from the curriculum is any mention of the value of (some would say necessity) of having mindfulness practice in addition to eating mindfully. Most people will be able to practice mindful eating better/easier if they also have a meditation practice…even if only for 10 min.” “It was clearly explained, but I would say missing some info about mindfulness in general…what it is, why do it, benefits, prior to explaining what mindful eating is.”
Detailed	“It was laid out in an organized way, with all the needed component (i.e., cookbook/recipes) in the appendices.” “[The curriculum had] a concise plan of time, attention and details of each section.” “…this is detailed and compassionate. I enjoy the group interaction.”

## Data Availability

Data are available upon request.

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
