# Peer review of "Development of Master Chef: A Curriculum to Promote Nutrition and Mindful Eating among College Students"

_ijerph, 2024, doi:10.3390/ijerph21040487_

Round 1

Reviewer 1 Report

Comments and Suggestions for Authors

Dear Sir / Madam,

Please find below my comments and suggestions following the review of the study titled ‘Development of Master Chef: A Curriculum to Promote Nutrition and Mindful Eating Among College Students’.

This study addresses an important topic concerning nutrition and mindful eating among college students, which is highly relevant given the increasing prevalence of unhealthy eating habits in this demographic. However, it is essential to note that the presentation of the study requires considerable revision. The manuscript lacks clarity which makes it difficult to comprehend. Some sections are described in too much details while other section lack adequate description. Suggestions to improve the manuscript have been presented below:

1.     Abstract:

·       The authors are requested to rectify language errors in the Abstract. For eg:

a.      Two independent researchers coded qualitative data and underwent inductive thematic analysis. This phrase implies that the researchers underwent inductive thematic analysis.

b.     Reviewers (N=7) were experts in the fields of nutrition, psychology, and mindful eating, respectfully. I think the authors mean to use the word ‘respectively’.

2.     Introduction:

·       The 'Introduction' section is long and repetitive. The authors are requested to shorten it.

3.     Materials and Methods:

·       The methods used by the authors are unclear. It would be helpful if the authors present methods in the form of a flow diagram.

4.     Results:

·       The authors mention that 110 experts were enrolled in the course – does this mean that 110 experts had to complete the course or had to review the course?

·       The authors say their goal was to recruit 20 reviewers but only 7 out of 110 participated. Why did the authors choose the number 20. Since less than half that number completed the review, was this adequate for a suitable review of the course?

·       Did the reviewers have access to the course content or only to the curriculum as presented in Table 1?

5.     Discussion:

·       The authors present how other nutrition education programs have been reviewed by experts and accepted as feasible to incorporate. If there are already programs which have been reviewed and which have been found to be acceptable, what is the reason for the authors to create a new program? Why is this new program better than the others?

·       The “Discussion’ section seems to focus more on what expert reviewers have said about other courses/programs than what the expert reviewers have said about the Master Chef course. The authors need to work to rectify this.

Comments on the Quality of English Language

 Moderate editing of English language required

Author Response

Hello,

Thank you for your suggestions. I was unable to upload my point-by-point revisions. I have copy and pasted the table below. Again, thank you for your time.

All the best,

Kayla

Reviewer One

Suggestions

Response

The authors are requested to rectify language errors in the Abstract. For eg:

a.      Two independent researchers coded qualitative data and underwent inductive thematic analysis. This phrase implies that the researchers underwent inductive thematic analysis.

b.     Reviewers (N=7) were experts in the fields of nutrition, psychology, and mindful eating, respectfully. I think the authors mean to use the word ‘respectively’

Thank you for this suggestion. The grammar in the abstract has been updated to the following:

a.              “Two independent researchers coded qualitative data, which then underwent inductive thematic analysis.”

b.              “Reviewers (N=7) were experts in the fields of nutrition, psychology, and mindful eating, respectively”

2.     Introduction:

·       The 'Introduction' section is long and repetitive. The authors are requested to shorten it.

Thank you for your suggestion. The introduction is now more concise and authors have removed repetitious information, specifically information regarding the relationship between mental health and diet quality. In addition, only pertinent information regarding other interventions has been included.

3.     Materials and Methods:

·       The methods used by the authors are unclear. It would be helpful if the authors present methods in the form of a flow diagram.

Thank you for your suggestion. A flow chart has been added for clarity within the methods.

4.     Results:

·       The authors mention that 110 experts were enrolled in the course – does this mean that 110 experts had to complete the course or had to review the course?

·       The authors say their goal was to recruit 20 reviewers but only 7 out of 110 participated. Why did the authors choose the number 20. Since less than half that number completed the review, was this adequate for a suitable review of the course?

·       Did the reviewers have access to the course content or only to the curriculum as presented in Table 1?

Thank you for your suggestion. 

It has now been clarified in the methods, rather than the results, that 110 individuals were enrolled in the online mindful eating course, which was used as a platform for recruitment efforts. Authors aimed to recruit at least 20 reviewers based on previous research conducted by Hagaman & Wutich. It has been clarified in the methods that reviewers were asked to read the entirety of the curriculum.The limitations now discuss how the authors were close in reaching the goal of 20 participants, but that a majority of those who attempted the survey had incomplete data sets. This suggests that critically reviewing a curriculum is time intensive and may be difficult for those working in the healthcare or academic field. 

Despite these challenges, saturation was reached likely due to the authors using purposeful sampling during recruitment. There are limited mindful eating experts who are jointly skilled in fields of nutrition, psychology, and program development, allowing for the current sample to provide rich data regarding feasibility of the program, despite low recruitment. Also of note, similar programs, such as Project Stride, Eat Right, Future Bright, and Eating Smart, Being Active, as described in the discussion have comparable sample sizes, ranging from five to nine experts in during an expert review. 

5.     Discussion:

·       The authors present how other nutrition education programs have been reviewed by experts and accepted as feasible to incorporate. If there are already programs which have been reviewed and which have been found to be acceptable, what is the reason for the authors to create a new program? Why is this new program better than the others?

·       The “Discussion’ section seems to focus more on what expert reviewers have said about other courses/programs than what the expert reviewers have said about the Master Chef course. The authors need to work to rectify this.

Thank you for your suggestions. Despite previous research efforts finding adolescent programming effective, there are limited attempts to assess college-aimed programming through use of expert review. Undergraduates uniquely face alarming levels of perceived stress and poor diet quality, requiring theory-based programming that promotes self-efficacy to overcome college-specific barriers. This is done through providing relevant tools for students (nutrition education, culinary skills and mindfulness).  Master Chef is unique in addressing these barriers, while also being vetted through experts. 

The discussion has now been clarified in discussing how expert reviewers of the Master Chef curriculum relate to the outcomes of other curriculums using an expert review..

Reviewer 2 Report

Comments and Suggestions for Authors

Healthy eating is one of the current societal challenges and obviously the paper can have significant contribution to this field, but there are gaps, which should be taken into consideration.

The abstract style should be changed. The abstract must be more concise. Please note that the part of the expert review could be more synthetic and obviously raise the main points. 

The introduction style also should be changed. Unfortunately, the introduction is not clearly structured in order to have a better view of the specific objective of the study. Please delete repetitions, I would suggest also to simplify the sentence structure, because the paper can be interesting not only for researches but also for other stakeholders. 

Material and method section also should be revised and changed. It lacks of specific information. Please expand the subsection of data analysis methods, providing more details on how qualitative and quantitative data have been analyzed. Please provide a clear information which types of descriptive statistics were used and how thematic analyses of qualitative data were conducted.

Provide a brief explanation of low expert participation compared to the number contacted. You may include some possible reasons for low participation and discuss how this might affect the reliability of the results. In my opinion the number of experts is too limited.

Discuss whether the scores might be affected by low number of expert or other factors. 

Expand the discussion on study limitations, including low expert participation, the possible influence of experts on results. There is a need of additional program validation tests. 

Comments on the Quality of English Language

Minor editing is needed.

Author Response

Hello,

I appreciate your time in reviewing this manuscript. I was unable to upload files to this portal. I have copied my table of point-by-point revisions.

Again, thank you for your time. 

Reviewer Two

Suggestions

Response

Healthy eating is one of the current societal challenges and obviously the paper can have a significant contribution to this field, but there are gaps, which should be taken into consideration.

The abstract style should be changed. The abstract must be more concise. Please note that the part of the expert review could be more synthetic and obviously raise the main points. 

Thank you for your suggestion. The abstract has been modified to be more concise. 

The introduction style also should be changed. Unfortunately, the introduction is not clearly structured in order to have a better view of the specific objective of the study. Please delete repetitions, I would suggest also to simplify the sentence structure, because the paper can be interesting not only for researchers but also for other stakeholders. 

Thank you for your suggestion. The introduction is now more concise and repetitious information has been removed, specifically information regarding the relationship between poor mental health and diet quality. Sentences and grammar have also been corrected throughout.

Material and method section also should be revised and changed. It lacks of specific information. Please expand the subsection of data analysis methods, providing more details on how qualitative and quantitative data have been analyzed. Please provide a clear information which types of descriptive statistics were used and how thematic analyses of qualitative data were conducted.

Thank you for your suggestion. The analysis section has been expanded upon, now describing the type of descriptive statistics used (frequency distribution and central tendency methods). The authors have also provided examples of what variables were assessed using each method. The process of inductive thematic analysis is now described in more depth as well.

Provide a brief explanation of low expert participation compared to the number contacted. You may include some possible reasons for low participation and discuss how this might affect the reliability of the results. In my opinion the number of experts is too limited.

The limitations now discuss how the authors were close in reaching the goal of 20 participants, but that a majority of those who attempted the survey had incomplete data sets. This suggests a limitation of using expert review is that it is time intensive and may be difficult for those working in the healthcare or academic field. Authors used purposeful sampling during recruitment based on the current subject matter. There are limited mindful eating experts who are jointly skilled in fields of nutrition, psychology, and program development, allowing for the current sample to provide rich data regarding feasibility of the program, despite low recruitment. Of note, similar program assessments, such as described in Project Stride, Eat Right, Future Bright, and Eating Smart-Being Active, had comparable sample sizes, ranging from five to nine experts in using expert review. 

Discuss whether the scores might be affected by low number of expert or other factors. Expand the discussion on study limitations, including low expert participation, the possible influence of experts on results. There is a need of additional program validation tests.

Thank you. The limitations have been expanded upon to describe the impact of a low recruitment sample on quantitative results assessing program feasibility. The authors agree that additional testing is needed for validation of program effectiveness.

Round 2

Reviewer 1 Report

Comments and Suggestions for Authors

The authors have done a good job incorporating the reviewer's suggestion.

With reference to this sentence in the Abstract, "Reviewers (N=7) were experts in the fields of nutrition, psychology, and mindful eating, respectively." the word respectively, too, seems to be unnecessary and the authors are requested to delete it.

I am still unclear about the role of the 110 experts who were enrolled in the course. What was expected from them? Did they submit any review? Did they complete the course? There is nothing mentioned about what the 100 reviewers contributed and this is confusing.

Also, if the goal was to recruit 20 reviewers, what was the justification for inviting 110 experts? The authors mention that review saturation was achieved with only 7 reviewers. However, this claim appears challenging to reconcile, since the authors themselves state references that 20 reviewers are recommended by previous research.

The authors are requested to improve the presentation of the methods and results section to improve clarity.

Comments on the Quality of English Language

Some English language editing is still required.

Author Response

Reviewer 1

Suggestion

Response

With reference to this sentence in the Abstract, "Reviewers (N=7) were experts in the fields of nutrition, psychology, and mindful eating, respectively." the word respectively, too, seems to be unnecessary and the authors are requested to delete it.

Thank you for your suggestion. This has been deleted.

I am still unclear about the role of the 110 experts who were enrolled in the course. What was expected from them? Did they submit any review? Did they complete the course? There is nothing mentioned about what the 100 reviewers contributed and this is confusing.

The authors apologize for any confusion. For clarity, 110 individuals were enrolled in the online mindful eating course platform which was run by a separate organization. With permission from the course administrators, the link for recruitment, consent form and coinciding expert review survey was posted to a discussion board, which would be available to all 110 individuals enrolled in the course. Participants in the mindful eating course were not required to complete the expert review as a part of course. This has been clarified in the methods.

Also, if the goal was to recruit 20 reviewers, what was the justification for inviting 110 experts? The authors mention that review saturation was achieved with only 7 reviewers. However, this claim appears challenging to reconcile, since the authors themselves state references that 20 reviewers are recommended by previous research.

Thank you for your suggestion. The 110 participants had equal access to the expert review link through the online mindful eating platform’s discussion board, and therefore were informally invited to complete the posted link. This has been updated in the manuscript.

Previous literature had informed the initial goal of 20 participants for the expert review to aim to reach saturation. Despite not meeting this participant recruitment, saturation was met using seven participants as seen through no new information or themes being observed in the data. As noted in the discussion, several similar studies using expert review to assess nutrition curricula have also reached saturation with similar sample sizes. It can be suggested that an increased sample size would not add additional insight to program refinement due to meeting saturation with the current sample. This has been updated in the discussion.

The authors are requested to improve the presentation of the methods and results section to improve clarity.

Thank you for your suggestion. The presentation of the methods has been modified to provide clarity to the reader. The previous ‘section 2.3’ has now been modified to, ‘2.3. Expert Review Recruitment’ and a new section has been added ‘2.4 Expert Review Survey’. Methods regarding recruitment have also been expanded upon. Section headers for the results have also been updated to provide more clarity.

Reviewer 2 Report

Comments and Suggestions for Authors

Can be published in a current form.

Comments on the Quality of English Language

Minor editing is needed 

Author Response

Thank you for your time and consideration of this manuscript. Grammar and language has been modified throughout for clarity.